# High-resolution crystal structure of a 20 kDa superfluorinated gold nanocluster

Claudia Pigliacelli [1,2], Angela Acocella [3], Isabel Díez[2], Luca Moretti [4], Valentina Dichiarante [1✉], Nicola Demitri [5], Hua Jiang[2], Margherita Maiuri[4], Robin H. A. Ras [2,6], Francesca Baldelli Bombelli [1], Giulio Cerullo [4], Francesco Zerbetto[3], Pierangelo Metrangolo [1,2✉] & Giancarlo Terraneo [1✉]

Crystallization of atomically precise nanoclusters is gaining increasing attention, due to the opportunity of elucidating both intracluster and intercluster packing modes, and exploiting the functionality of the resulting highly pure crystallized materials. Herein, we report the design and single-crystal X-ray structure of a superfluorinated 20 kDa gold nanocluster, with an $Au_{25}$ core coated by a shell of multi-branched highly fluorinated thiols ($SF_{27}$) resulting in almost 500 fluorine atoms, i.e., ($[Au_{25}(SF_{27})_{18}]^0$). The cluster shows a switchable solubility in the fluorous phase. X-ray analysis and computational studies reveal the key role of both intracluster and intercluster F···F contacts in driving $[Au_{25}(SF_{27})_{18}]^0$ crystal packing and stabilization, highlighting the ability of multi-branched fluorinated thiols to endow atomically precise nanoclusters with remarkable crystallogenic behavior.

[1] Laboratory of Supramolecular and Bio-Nanomaterials (SupraBioNanoLab), Department of Chemistry, Materials, and Chemical Engineering "Giulio Natta", Politecnico di Milano, via L. Mancinelli 7, 20131 Milano, Italy. [2] Department of Applied Physics, Aalto University School of Science, Puumiehenkuja 2, FI-00076 Espoo, Finland. [3] Dipartimento di Chimica "G. Ciamician", Università di Bologna, V. F. Selmi 2, 40126 Bologna, Italy. [4] IFN-CNR, Dipartimento di Fisica, Politecnico di Milano, 20133 Milano, Italy. [5] Elettra-Sincrotrone Trieste S.S. 14 Km 163.5 in Area Science Park, 34149 Basovizza, Trieste, Italy. [6] Department of Bioproducts and Biosystems, Aalto University School of Chemical Engineering, P.O. Box 16000, FI-00076 Espoo, Finland. ✉email: valentina.dichiarante@polimi.it; pierangelo.metrangolo@polimi.it; giancarlo.terraneo@polimi.it

Gold nanoclusters (AuNC) with atomically precise sizes represent a distinct class of nanomaterials that can be identified by a single chemical formula, $[Au_n(L)_m]^q$, and exhibit molecule-like electronic and optical properties[1,2]. Such a unique set of features in a nanoscale system has been rationalized as the result of the 'size quantization' of AuNC electronic band structure, giving rise to defined UV-visible absorption, fluorescence, and catalytic behavior[3]. Owing to these distinguished properties, AuNCs are turning promising in several high-end fields, including sensing, nanomedicine, energy conversion, and catalysis[4], and are steadily emerging as valuable building blocks in material and crystal design[5–8]. AuNC monodisperse size and structural accuracy rapidly led to their successful crystallization and to the determination of the first high-resolution crystal structure for $Au_{102}(L)_{44}$, obtained in 2007 by Kornberg and colleagues[9]. Since this seminal finding, several AuNC X-ray structures have been reported, with cores ranging from $Au_{18}$ to $Au_{133}$[2,10]. Among them, $Au_{25}(L)_{18}$ appears the most outstanding, as it is usually synthesized in high yields under mild conditions and exists in different charge states ($q = -1, 0, +1$), with charge-dependent properties[11].

To date, thiols (RSH) represent the most popular class of stabilizing ligands for AuNCs, owing to the strong S–Au bond[12]. In an attempt to introduce additional surface functionalities and broaden the range of possible AuNCs, thiols bearing azide moieties or pyridyl groups were recently used to stabilize novel $Au_{25}NC$, whereas N-heterocyclic carbenes have been employed to obtain nanoclusters with an $Au_{11}$ core[13–15]. Another emerging class of ligands for colloidal gold nanostructures is represented by highly fluorinated thiols[16,17]. Notably, the replacement of H by F in alkyl chains leads to enhanced hydrophobicity and stiffness, and promotes self-assembly, ordered stacking, and segregation between fluorinated and non-fluorinated moieties (i.e., fluorophobic effect)[18,19]. Although F⋯F contacts are generally weak and repulsive, in some cases they act collectively, giving rise to highly stabilizing interaction energies, and thus working as driving forces for crystal packing, or providing complementary stabilization, as already shown for small molecules in the solid state[20], and for more complex substrates, like polymers[21] and dendrimers[22]. Despite the established role of fluorinated compounds in both crystal engineering and functional materials design[23–25], very few examples of atomically precise metal nanoclusters having a high content of fluorine atoms in their ligand-shells have been reported, to now. Specifically, crystal structures have been obtained for Ag and intermetal clusters stabilized by partially fluorinated arylthiols[26–31], and only four structures of AuNCs have been obtained using alkynyl ligands functionalized with $CF_3$ units[32–34]. However, in all the reported cases the role of fluorinated units, and thus of F⋯F contacts, in promoting the self-assembly of the nanoclusters was not fully addressed.

Herein we report the high-resolution single-crystal X-ray structure of a superfluorinated AuNC ($[Au_{25}(C_{20}H_{14}F_{27}O_4S)_{18}]$ and hereafter $[Au_{25}(SF_{27})_{18}]^0$) with the Au core coated by a shell bearing almost 500 fluorine atoms, showing molecule-like optical absorption, and switchable solubility in the fluorous phase. The spontaneous crystallization of this species from a bulk AuNC dispersion was driven by a bulky multi-branched highly fluorinated thiol ($C_{20}H_{14}F_{27}O_4SH$ and hereafter $F_{27}SH$, Fig. 1) employed as stabilizing ligand, whose molecular scaffold has a strong tendency to yield crystalline materials[35]. Our hypothesis was that, by binding the gold core, $F_{27}S$-ligand would achieve an optimal spatial distribution that could favor the F⋯F contacts as key supramolecular tools for the assembly of fluorinated AuNC in the solid-state. Moreover, we investigated the effect of highly fluorinated ligands on the spectroscopic behavior of the gold core.

## Results and discussion

**F⋯F Interactions in the $F_{27}SH$ ligand.** Our studies began with a computational investigation aimed at uncovering the nature of F⋯F contacts occurring in $F_{27}SH$, both intra and intermolecularly, to assess their possible impact on the supramolecular and functional features of $F_{27}S$-stabilized AuNC. In detail, through a bond order (BO) analysis, i.e., evaluation of the shared electron density between atoms, and a potential energy surface mapping between interacting fluorine atoms, we unambiguously characterized F⋯F contacts on previously reported $F_{27}SH$ crystallographic data (Fig. 1a)[35]. As shown in Fig. 1b, F⋯F interactions can be distinguished according to a different degree of delocalization (Supplementary Method 3), displaying a negative, stabilizing energy when F⋯F contacts are longer than 2.66 Å and the corresponding calculated BO is lower than 0.07 au. Differently, for interatomic distances shorter than 2.66 Å (higher than 0.07 au), the resulting interaction energy is positive and has, therefore, a destabilizing nature.

Thus, both intra- and inter-branches, as well as inter-thiol interactions, due to their relative space arrangement in the crystalline structure, primarily show an attractive nature and, although independently weak, they are numerous and act additively (Fig. 1b, c), contributing with an overall stabilization of about 15.5 kcal/mol. This picture allows to describe non-covalent F⋯F contacts as a network of weak but locally stabilizing interactions, assisting $F_{27}SH$ crystallization.

Further inspection of the electron density distribution, based on the non-covalent interactions, NCI, analysis method[36] corroborates the BO analysis findings, confirming the overall stabilizing nature of F⋯F contacts occurring in the $F_{27}SH$ molecule. Different local bonding regions, with their relative strength, can be readily visualized in real space by plotting the calculated reduced electron density gradient (RDG) as a function of electron density ($\rho$), and classified as attractive or repulsive, according to the sign of the second density Hessian eigenvalue sign ($\lambda_2$; Supplementary Method 3)[37]. In particular, as shown in Fig. 1d, the NCI plot derived on the $F_{27}SH$ crystallographic data shows the presence of steep peaks at low density values, indicating van der Waals weakly attractive interactions between fluorine contacts. Moreover, NCI three-dimensional iso-surface plots (Fig. 1e, f) calculated, respectively, on a single thiol molecule and in two nearby thiols extracted from the crystal structure, provide a three-dimensional mapping of F⋯F interactions, which agrees with the BO distribution analysis, confirming the overall stabilizing nature of the F⋯F contacts present in the $F_{27}SH$ molecule, and their role in driving its crystallization.

**$Au_{25}(SF_{27})_{18}$ synthesis and crystallization.** Given $F_{27}SH$ self-assembly features, we reasoned that such a unique molecule could represent an optimal stabilizing ligand to devise AuNC with atomic precision and crystallization ability (Fig. 2a). A slightly modified Brust reaction, performed using $HAuCl_4$ and thiol $F_{27}SH$ in 1:3 molar ratio, afforded fluorinated AuNC (F-AuNC), which were easily dispersed in 1,1,1,3,3-pentafluorobutane (solkane). After two months, the sample, stored at room temperature, presented the formation of dark crystals that were scarcely soluble in a partially fluorinated solvent like solkane, but dissolved well in a fluorous medium such as perfluorooctane (PFO). Interestingly, crystals in PFO gave a light green solution, whereas the starting crude product in solkane had a dark brown color (Fig. 2a and Supplementary Fig. 5).

UV–visible (UV–Vis) studies performed on F-AuNC in solkane and redissolved crystals in PFO highlighted relevant differences between the two samples. As shown by Fig. 2c and Supplementary Fig. 6a, b, UV–Vis spectrum of the crude synthesis product showed broad peaks at 395, 453, 546, and 650 nm, which were characteristic of charged $[Au_{25}(SR)_{18}]^-$ species[3,38]. Additionally, the presence of larger Au nanoparticles

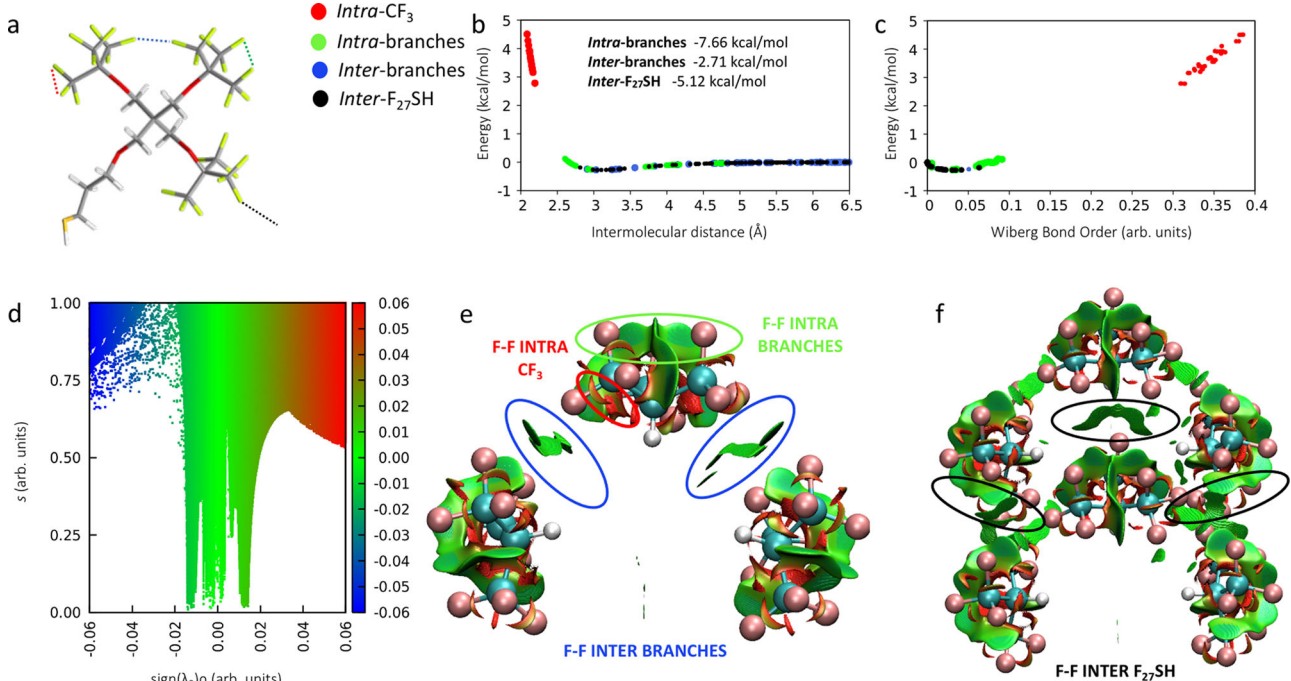

**Fig. 1 DFT characterization of F⋯F interactions in the F$_{27}$SH crystal structure. a** Molecular structure of F$_{27}$SH and graphical representation of *intra*- and *inter*-molecular F⋯F interactions in F$_{27}$SH. Atom color code: C, gray; O, red; F, light yellow; S, dark yellow and H, white. F⋯F interactions color code: red, *intra*-CF$_3$; green, F⋯F *intra*-branches; blue, *inter*-branches; and black, F⋯F *inter*-F$_{27}$SH; **b**, **c** F⋯F intermolecular interaction energy calculated at M062x/aug-cc-pvtz level of theory with GD3 Grimme's empirical dispersion corrections and Wiberg bond orders; the inset of **b** reports the total interacting energy (kcal/mol) for every F⋯F interacting pair; **d** non-covalent interactions map calculated at the same level of theory of **b** and **c** showing *intra* and *inter* non-covalent interacting regions between the branches of two nearby thiol molecules. The reduced density gradient is plotted *versus* the sign of the second density Hessian eigenvalue multiplied by the density itself (sign($\lambda_2$)$\rho$); the conventional color code applied uses blue to indicate attractive (negative values of sign($\lambda_2$)$\rho$) interactions, red for repulsive (positive values of sign($\lambda_2$)$\rho$) interactions and green for weak van der Waals interactions (sign($\lambda_2$)$\rho$ close to zero); **e**, **f** three-dimensional iso-surface plots of non-covalent interactions (isovalue 0.8) calculated in a single thiol molecule and in two nearby thiols visualizing interactions as green surfaces for weak van der Waals, red surfaces for non-bonding repulsive steric interactions. Red, green, blue, and black colors correspond to the classification of F⋯F interactions indicated in this work.

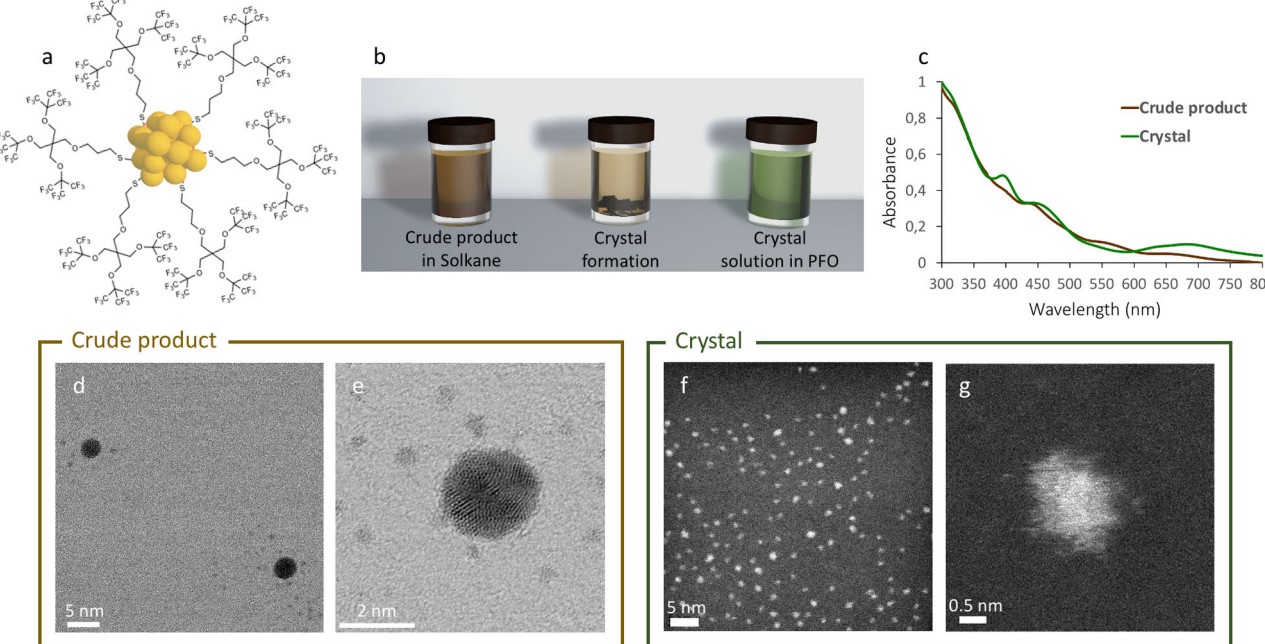

**Fig. 2 Synthesis of AuNC stabilized by F$_{27}$SH and crystallization of [Au$_{25}$(SF$_{27}$)$_{18}$]$^0$. a** Cartoon representation of AuNC stabilized by F$_{27}$SH thiol. For the sake of clarity, only 6 F$_{27}$S-ligands have been reported; **b** schematic representation of crystal formation in solkane solution and colorimetric change upon their dissolution in PFO; **c** UV–Vis spectra of crude product in solkane and crystals redissolved in PFO; **d**, **e** STEM images of crude product showing the presence of small clusters and bigger AuNPs; **f**, **g** STEM images of redissolved crystals solution showing the homogeneous presence of small clusters.

(AuNPs; >2 nm) was revealed by scanning transmission electron microscopy (STEM) characterization, as shown in Fig. 2d, e and Supplementary Fig. 7. Upon crystal formation and subsequent dissolution in PFO, UV–Vis profile of the resulting sample displayed an absorption band at 400 nm, a shoulder at 458 nm, and another broad peak at 690 nm, in accordance to what was previously reported for neutral $[Au_{25}(SR)_{18}]^0$ species (Fig. 2c and Supplementary Fig. 6c, d)[13]. Notably, a significant increase of peak sharpness could also be observed, indicating selective crystallization of $[Au_{25}(SF_{27})_{18}]^0$ species and high purity of the crystal, which is confirmed by STEM micrographs in Fig. 2f, g. These findings suggest the initial formation of negatively charged $[Au_{25}(SF_{27})_{18}]^-$ clusters, together with few larger AuNP, in the crude product, followed by slow oxidation to the neutral charge state, which results in the spontaneous formation of macroscopic crystals. Oxidation of F-AuNC is also revealed by the different color of the two species in solution, a colorimetric effect that has been previously reported for thiol-stabilized AuNC[13]. Moreover, this process matches well with the different solubility observed for F-AuNC species before and after crystallization, with neutral $[Au_{25}(SF_{27})_{18}]^0$ clusters being less polar and more fluorophilic than the negatively charged ones. This gives the opportunity to switch cluster solubility between two immiscible phases, by controlling its oxidation state.

The composition of crystalline F-AuNC was further probed by high-resolution matrix-assisted laser desorption ionization time-of-flight (HR-MALDI-TOF) mass spectrometry. Given $[Au_{25}(SF_{27})_{18}]^0$ molecular weight (20464.9 g/mol) and neutral charge, a parent cluster peak was expected at 20464.9 m/z. Although mild laser power conditions were employed, the predicted parent peak could not be found. However, obtained data showed an intense peak centered at 16168 m/z (see Supplementary Fig. 8), corresponding to a first $[Au_{25}(SF_{27})_{18}]^0$ fragment, i.e., $Au_{25}(SF_{27})_{13}$, followed by a fragmentation pattern indicating the gradual loss of one Au atom and one thiol molecule by the first fragment (1060 m/z), proving the instability of the analyzed sample under laser irradiation. Further, $^1H$ and $^{19}F$ nuclear magnetic resonance analyses also confirmed that the structure of $[Au_{25}(SF_{27})_{18}]^0$ remains intact in solution (see Supplementary Figs. 9 and 10).

**Optical properties of $[Au_{25}(SF_{27})_{18}]^0$.** F-AuNCs optical properties and their energy relaxation dynamics were probed by femtosecond transient absorption (TA) spectroscopy. Figure 3 reports the differential absorption ($\Delta A$) map of $[Au_{25}(SF_{27})_{18}]^0$ PFO solution. Following photoexcitation, instantaneous broad positive spectral features dominating the $\Delta A$ map in the visible and the near-IR were observed (Fig. 3a), which were assigned to excited state absorption (ESA) bands from the Au core to higher energy states following an ultrafast (below our temporal resolution) relaxation from the interband transition to the lowest energy (HOMO-LUMO) $Au_{13}$ core transition. By monitoring the evolution of selected time traces at specific probe wavelengths, an ultrafast (sub-picosecond) decay of the transient $\Delta A$ signal was detected, followed by a build-up of the ESA band within few tens of picoseconds in the blue part of the probe spectrum (Fig. 3b). To simultaneously fit the dynamics at different wavelengths, we performed a global analysis employing a 4-state sequential model (see details in the "Methods" section). The evolution associated spectra (EAS) together with their exponential decay times are reported in Fig. 3c. The first EAS decay with a time constant $\tau_1 = 442$ fs is associated to the internal conversion (IC) from the $Au_{13}$ core to ligand-localized electronic semiring states. This decay time is faster than what previously reported on $Au_{25}$ clusters (1.2 ps[39] or 0.7 ps[40]), possibly due to a distortion induced

by the presence of the bulky $F_{27}S$-ligands promoting faster IC, as proposed for an $Au_{24}Pd$ cluster when Pd is located at the ligand-site[40]. The second EAS showed, with respect to the first one, a red shift of the ESA peak from 570 to 590 nm, together with a decay all over the spectrum. The evolution from the second to the third EAS occurred with a time constant $\tau_2 = 11$ ps, which was assigned to nuclear-electron reorganization following the localization of the ligands excitation, which led to the formation of charge transfer (CT) states[41]. This time constant is slower with respect to similar AuNC[39], due to the larger inertia associated with the fluorinated ligands rearrangement. The third component with $\tau = 212$ ps showed a blue shift of the ESA band in the 450–650 nm region. This decay was attributed to non-radiative relaxation from semiring states. Finally, the fourth component was a non-decaying plateau with a time constant longer than our temporal range ($\tau_4 > 1$ ns) and captured the slow radiative recombination from the long-lived CT states centered at 1200 nm[39]. The energy level scheme and the relaxation processes are summarized in Fig. 3d.

Finally, we noticed the presence of coherent oscillations superimposed over the transient dynamics, mainly visible at short probe wavelengths (Fig. 3e). Figure 3e shows the oscillatory signal observed at 350 nm probe wavelength after removing the slowly varying decay component. Following Fourier analysis, two low-frequency modes at 60 and 140 $cm^{-1}$ can be seen. The former one has previously been attributed to the acoustic phonons of the $Au_{13}$ core[39,42], with a dephasing time matching the core IC process (about 400 fs). The second higher frequency mode was connected to the semiring region[39], thus involving indirectly the fluorinated ligands. These oscillations were much more persistent and were related to the 11 ps time decay for the relaxation in the semiring states. Thus, the analysis of these coherent oscillations highlighted the key role and the impact of the bulky fluorinated $F_{27}S$-ligands on the $[Au_{25}(SF_{27})_{18}]^0$ optical properties.

**High-resolution single-crystal X-ray structure and role of F···F interactions in the crystal packing.** Single-crystal X-ray studies were performed to confirm the chemical composition inferred by UV–Vis and HR-MALDI experiments and get insights into the $[Au_{25}(SF_{27})_{18}]^0$ structure. Dark prismatic crystals were grown by slow evaporation of a diluted solkane solution (two months, room temperature). F-AuNC crystallized in the triclinic space group P-1 as solkane solvate, having one central gold atom ($Au_{cent}$) at the inversion center (Supplementary Fig. 11). The $Au_{25}$ kernel adopted an icosahedral arrangement, with $Au_{cent}$ surrounded by 12 inner gold atoms ($Au_{inn}$). Au–Au bond distances between the central Au atom and $Au_{inn}$ atoms spanned from 2.7786(9) to 2.7998(7) Å, with a slightly longer average $Au_{inn}$–$Au_{inn}$ bond length (2.934 Å, Supplementary Fig. 12 and Supplementary Table 2).

Each $Au_{inn}$ atom was further bound to three outer gold atoms ($Au_{out}$), with $Au_{inn}$–$Au_{out}$ average bond distance of 3.141 Å (Supplementary Fig. 13). In the overall star-shaped architecture of the gold core (Fig. 4a), Au–Au bond lengths followed the order: $Au_{cent}$–$Au_{inn}$ > $Au_{inn}$–$Au_{inn}$ > $Au_{inn}$–$Au_{out}$, as previously reported for other icosahedral $Au_{25}$ NCs[43–45].

The coordination sphere of the $Au_{25}$ core was completed by a series of S–$Au_{out}$–S–$Au_{out}$–S staple motifs (Fig. 4b), with the two terminal sulfur atoms covalently bound to one gold atom of the $Au_{13}$ kernel and one $Au_{out}$ atom (average S–Au bond length: 2.323 Å; mean S–Au–S angle: 173°; average S–S–S angle: 110°). This structural motif was repeated six times around the gold core, creating the anchoring points for the 18 $F_{27}S$-ligands. All S–$Au_{out}$–S–$Au_{out}$–S motifs adopted an almost planar arrangement with the three sulfur atoms lying on a plane formed

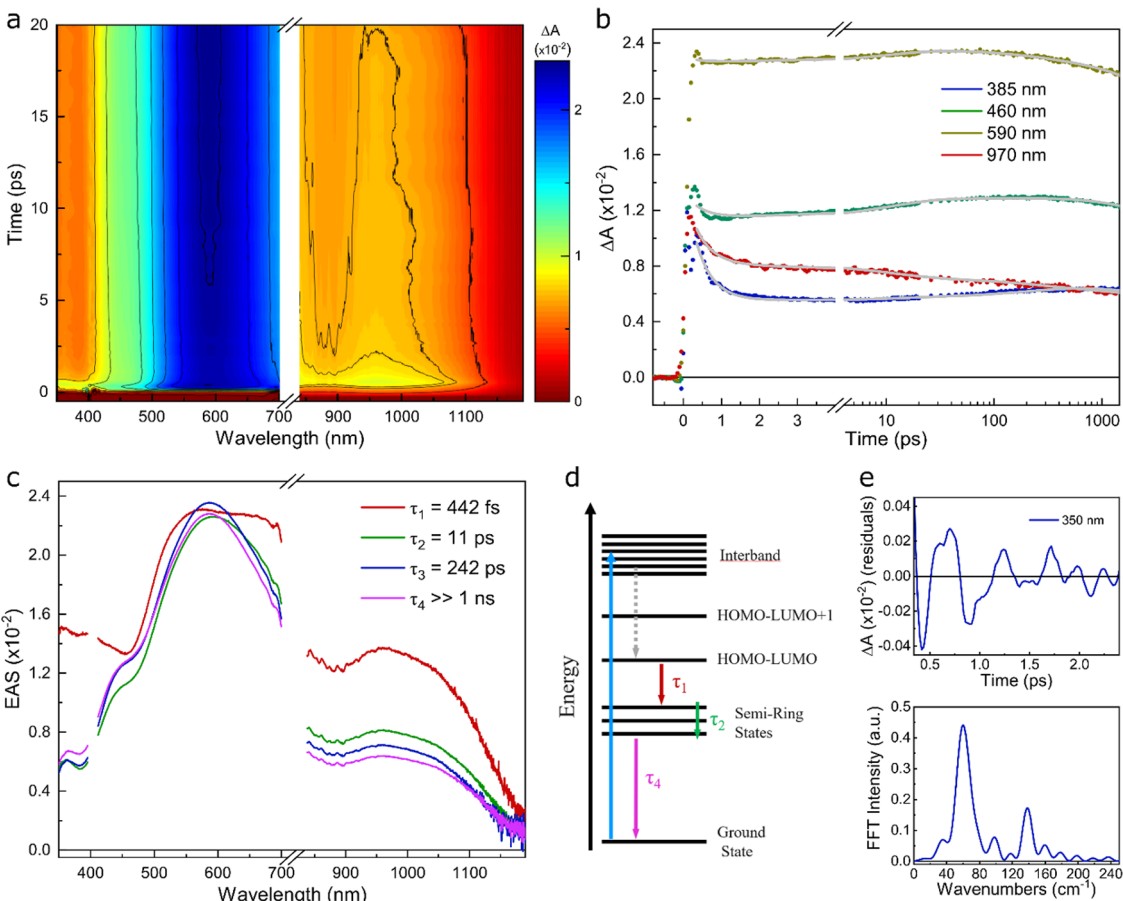

**Fig. 3 Ultrafast spectroscopy of [Au$_{25}$(SF$_{27}$)$_{18}$]$^0$. a** $\Delta A$ map of the crystal solution in PFO as a function of probe delay and probe wavelength. **b** $\Delta A$ time traces selected at specific probe wavelengths. **c** EAS spectra obtained from global analysis fit. **d** Model of electronic energy relaxation following 400 nm pump excitation. **e** Extracted coherent oscillations from $\Delta A$ time trace measured at 350 nm probe wavelength, after subtraction of a bi-exponential decay and its corresponding Fourier transform spectrum.

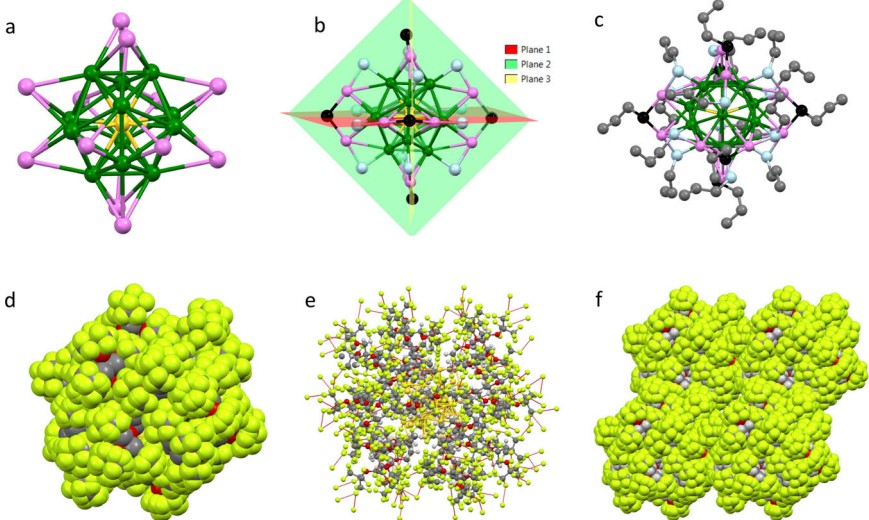

**Fig. 4 Crystallography of [Au$_{25}$(SF$_{27}$)$_{18}$]$^0$ (Mercury CSD 4.2.0). a** Au$_{25}$ core. Color code: Au$_{cent}$ in yellow, Au$_{inn}$ in green, Au$_{out}$ in pink; **b** Au$_{25}$S$_{18}$ unit, highlighting the six S–Au$_{out}$–S–Au$_{out}$–S staple motifs and the three planes they lay on. Sulfur atoms are colored in light blue (terminal) and black (bridged), respectively; **c** Au$_{25}$S$_{18}$ unit with C$_3$ alkyl portion of each thiol. The carbon atoms are colored in gray; **d** representation of one **[Au$_{25}$(SF$_{27}$)$_{18}$]$^0$** NC showing the fluorine masking of the gold core; **e** representation of one **[Au$_{25}$(SF$_{27}$)$_{18}$]$^0$** NC where the inter-cluster F⋯F contacts are shown as red dotted lines; **f** overall crystal packing of four **[Au$_{25}$(SF$_{27}$)$_{18}$]$^0$** NCs, view along the crystallographic *b* axis. **a**–**c**, **e** in ball-and-stick model, **d**, **f** in space-fill model.

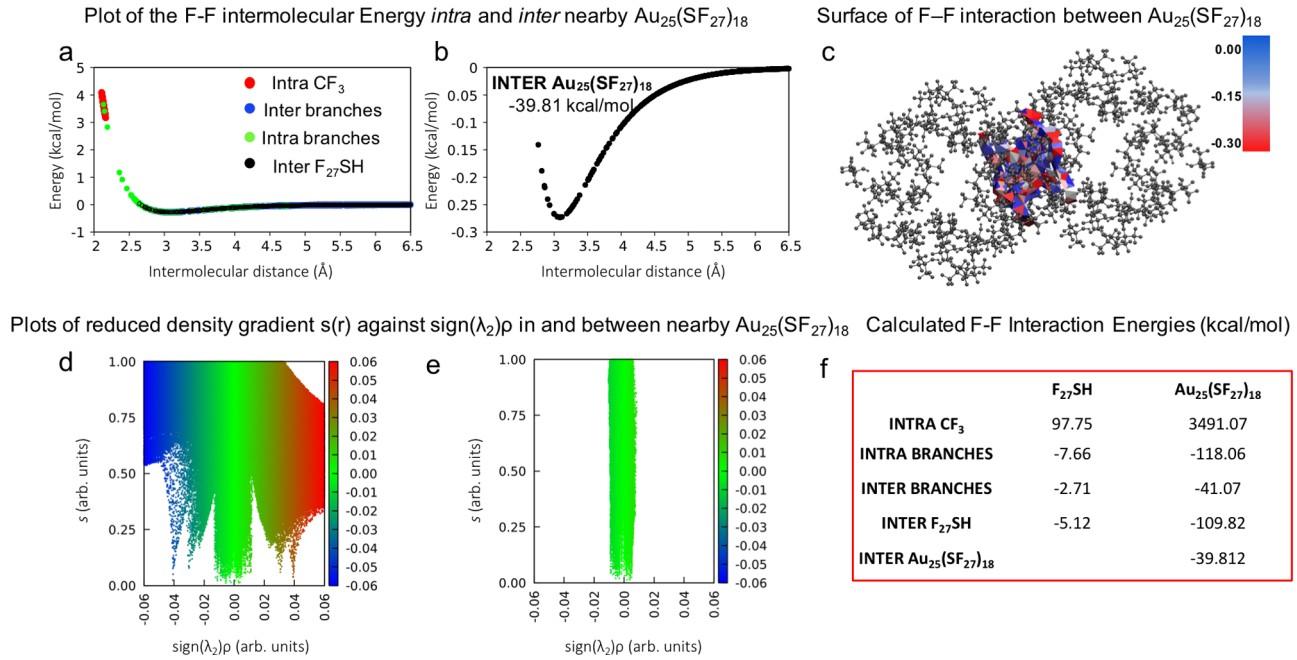

**Fig. 5 DFT characterization of F···F interactions in the [Au$_{25}$(SF$_{27}$)$_{18}$]$^0$ crystal structure. a, b** F···F Intermolecular interaction energy calculated at M062x/aug-cc-pvtz level of theory with GD3 Grimme's empirical dispersion corrections as a function of *intra-* and *inter-*cluster F···F distances between two cluster F$_{27}$S- ligands. Color code: red, *intra-*CF$_3$; green, F···F *intra-*branches; blue, F···F *inter-*branches; and black, F···F *inter-*F$_{27}$S-interactions; the inset text box of **b** reports the total interaction energy (kcal/mol) for every F···F interacting cluster pair; **c** surface representation of F···F intermolecular interaction energy in the *inter-*cluster region; color code: −0.30 (red) < −0.15 (white) < 0.00 (blue); **d** non-covalent interaction map visualized by using promolecular densities showing *intra-* and *inter-* non-covalent interacting regions of two nearby clusters. The reduced density gradient is plotted versus the sign of the second density Hessian eigenvalue multiplied by the density itself (sign($\lambda_2$)$\rho$); color code: blue for attractive (negative values of sign($\lambda_2$)$\rho$) interactions, red for repulsive (positive values of sign($\lambda_2$)$\rho$) interactions and green for weak van der Waals interactions (sign($\lambda_2$)$\rho$ close to zero); **e** magnification of an area of **d** for *inter-*cluster F···F interactions; **f** table of total interacting energy values (kcal/mol) calculated for every F···F interacting pair.

by Au$_{cent}$ and the two Au$_{out}$ atoms belonging to the staple itself (Fig. 4b). The staple motifs geometrical parameters were similar to those reported for non-fluorinated Au$_{25}$(SR)$_{18}$ systems[31–33], suggesting that the presence of bulky and highly fluorinated ligands did not affect much the Au$_{25}$ cluster architecture. The non-distorted planar geometry for the S–Au$_{out}$–S–Au$_{out}$–S motif has already been observed for uncharged icosahedral Au$_{25}$ nanoclusters[46], confirming the neutral charge state for [Au$_{25}$(SF$_{27}$)$_{18}$]$^0$ in the solid state, which was further corroborated by the absence of a counterion in the crystal lattice.

In contrast to the highly ordered Au$_{25}$ core, the surrounding fluorinated ligands show a significant degree of flexibility, and thus, disorder. Interestingly, the C$_3$ alkyl portion of each thiol molecule adopts a rather extended conformation, creating a sort of hydrocarbon layer between the Au core and the external fully CF$_3$-functionalized shell (Fig. 4c). Such a conformation is stabilized by several intramolecular hydrogen bonds, involving H atoms of CH$_2$ groups as electron density acceptor sites, and either O or F atoms as donors. No O···H contacts are detected between adjacent thiol fragments. Strikingly, an extended network of intermolecular F···H and F···F contacts is present in the ligands outer shell, driving the spatial distribution of its 162 CF$_3$ groups into a homogeneous fully fluorinated coating, which almost completely masks the Au cluster (Fig. 4d). This organization confirms the key role of F···F interactions in properly orienting the CF$_3$ groups between adjacent thiol ligands (F···F distances in the 2.7–2.9 Å range), and thus optimizing the packing forces for [Au$_{25}$(SF$_{27}$)$_{18}$]$^0$ in the solid state. Finally, the NC globular shape has an average radius of 16 Å, and its fluorinated corona of 486 F atoms establishes a continuum of F···F contacts that promote the supramolecular assembly of adjacent spherical NCs (Fig. 4e, f).

Data mining the Cambridge Structural Database[47] (CSD 2021.01) revealed only one crystal structure of a cluster having more than 200 F atoms (CCDC code: GUGSIN[30]). This structure has a metal core of 112 silver atoms capped with 51 3,5-bis(trifluoromethyl)phenylacetylide ligands, which leads to a total number of fluorine atoms of 306. If the search is restricted to AuNCs, only four crystal structures have been reported with fluorinated ligands and are the following: UKOCAB[33] (Au67 and F192), UKOCEF[33] (Au106 and F240), BUWJEL[34] (Au110 and F144), and ZIWXIP[32] (Au25 and F108) respectively. Therefore, to the best of our knowledge, [Au$_{25}$(SF$_{27}$)$_{18}$]$^0$ appears to be the nanoobject with the highest number of fluorine atoms ever solved at atomic level, by single-crystal X-ray diffraction (see section S.10 for more details). In all of the previously reported fluorinated AuNCs, (CF$_3$)$_2$-phenylacetylide units were the fluorinated moieties used to decorate the outer shell of the AuNCs. However, the rationale behind the use of fluorinated groups in the ligand design and the role of F···F interactions in the NC self-assembly process were not addressed.

The role of F···F interactions as packing forces for the crystallization of [Au$_{25}$(SF$_{27}$)$_{18}$]$^0$ was further confirmed by quantum chemical studies. Figure 5a shows F···F contacts within F$_{27}$S- ligands and between them in [Au$_{25}$(SF$_{27}$)$_{18}$]$^0$. Despite the presence of a small number of *intra-*branches contacts that fall in the repulsive region, the overall energy contribution is stabilizing, suggesting an additive contribution of F···F contacts to the formation of [Au$_{25}$(SF$_{27}$)$_{18}$]$^0$ crystals. As shown by Fig. 5b, all the *inter-*cluster F···F interactions have an attractive and thus stabilizing nature (see Fig. 5f for interaction energies). Figure 5d, e shows NCI analysis that provides further insight in the F···F interactions within and between [Au$_{25}$(SF$_{27}$)$_{18}$]$^0$ clusters, wherein

green indicates weak van der Waals contacts, red non-bonding repulsive steric interactions, and blue attractive interactions. Therefore, both crystallographic and computational studies confirm the two-fold role of $F_{27}S$-ligands, which efficiently stabilize AuNCs and also drive their crystallization through F⋯F interactions into highly pure single crystals.

In Summary, we have reported the design, synthesis, and crystallization of a superfluorinated atomically precise Au nanocluster bearing 486 F atoms, which represents the most fluorinated nanoobject ever described in the crystalline state. The crude synthetic product presented $[Au_{25}(SF_{27})_{18}]^-$ species, which spontaneously oxidized yielding, in the form of dark crystals, $[Au_{25}(SF_{27})_{18}]^0$ nanoclusters. Relaxation dynamics on $[Au_{25}(SF_{27})_{18}]^0$ was probed via femtosecond TA spectroscopy, which revealed the presence of core and ligand-localized electronic states. TA experiments found that core to semiring IC is 2-3 times faster than what previously reported for $Au_{25}$ clusters, while the ligand fluorinated chains cause a slower equilibration of the semiring states. High-resolution single-crystal X-ray diffraction studies highlighted the key role of F⋯F contacts in driving the crystallization of $[Au_{25}(SF_{27})_{18}]^0$. This was confirmed through BO and NCI computational studies, which clarified the attractive and additive nature of F⋯F contacts in both starting $F_{27}SH$ thiol and $[Au_{25}(SF_{27})_{18}]^0$ nanoclusters, while they are usually considered repulsive.

Our findings reveal the ability of a multi-branched super-fluorinated thiol to stabilize atomically precise nanoclusters and drive their crystallization through F⋯F interactions. Notably, the use of fluorinated ligands not only provides the clusters with valuable self-assembly properties, but also endows them with additional features, such as solubility in the fluorous phase. The investigation of the reported clusters in phase switching catalysis in a biphasic solvent system is currently undergoing and will be reported elsewhere.

## Methods

**Electron microscopy**. High-resolution STEM measurements were carried out with a JEOL 2200FS double aberration-corrected FEG TEM/STEM, operating at 200 kV. Solutions were dropped onto a TEM grid (carbon films 400 mesh Au).

**Mass spectrometry**. An Autoflex II instrument from Bruker Daltoniks (Bremen, Germany) equipped with a UV/N2-laser (337 nm/100 lJ) was used to carry out MALDI analyses. 2,3,4,5,6-Pentafluorobenzoic acid dissolved in solkane was used as the matrix. The purified fluorinated nanoclusters and DCTB, both dissolved in solkane, were mixed in a 1:1 (v/v) ratio and applied on the stainless-steel target plate in 1 μL aliquots. The sample spot was dried in air at room temperature. The mass spectrum (4–20 kDa) was measured in linear positive-ion mode, typically performing 1500 scans, and Protein standard solution II (Bruker Daltonics) was used for the external molecular mass calibration.

**X-ray**. Data collections were performed at the X-ray diffraction beamline (XRD1) of the Elettra Synchrotron, Trieste (Italy). The crystals were dipped in NHV oil (Jena Bioscience, Jena, Germany) and mounted on the goniometer head with kapton loops (MiTeGen, Ithaca, USA). Complete datasets were collected at 100 K through the rotating crystal method. Data were acquired using a monochromatic wavelength of 0.700 Å, on a Pilatus 2 M hybrid-pixel area detector (DECTRIS Ltd., Baden-Daettwil, Switzerland). The structures were solved by the dual space algorithm implemented in the SHELXT code. Fourier analysis and refinement were performed by the full-matrix least-squares methods based on $F^2$ implemented in SHELXL (Version 2017/1)[48]. The Coot program was used for modeling[49]. Electron content of cavities has been estimated with the SQUEEZE routine of PLATON[50]. No ordered solvent molecules could be modeled in the asymmetric unit (ASU) of $[Au_{25}(SF_{27})_{18}]^0$, therefore not construable residual density have been squeezed (563 electrons in 8%—1278 Å$^3$—of the unit cell volume). The disordered solvent has been estimated as additional three solkane solvent molecules in the ASU (1,1,1,3,3-pentafluorobutane or 1,1,1,2,3,3,3-heptafluoropropane).

**Computational methods**. All quantum chemical calculations were performed by means of Gaussian16 suite of program[51], with the Minnesota 2006 hybrid meta exchange-correlation functional (M06-2X),[39,52] Grimme's GD3 Empirical dispersion corrections[53] were added to account for dispersion energy contribution, in

combination with the basis set superposition error (BSSE)[53]. Further details can be found in the Supplementary Method 3. NCI maps and the corresponding 3d isosurfaces were produced by means of the NCI4 plot program[54] run on the wavefunction calculated with Gaussian16 program.

**Spectroscopy**. UV–Vis spectra of fluorinated gold nanoclusters bulk solution in solkane and of $[Au_{25}(SF_{27})_{18}]^0$ crystals redissolved in PFO at the appropriate dilution were acquired at room temperature on an Agilent Cary 5000 UV–Vis spectrometer.

Transient Absorption (TA): Ultrafast TA experiments were performed using an amplified Ti:Sapphire laser (Coherent Libra), with 800 nm central wavelength, 1 kHz repetition rate, 4 mJ output energy, and 100 fs pulse duration. The 400 nm excitation pulses were generated by second harmonic generation (SHG) using a β-barium borate crystal. The white-light continuum (WLC) used for the broadband probe pulses was generated in two ways. A WLC spanning 350–700 nm was obtained by focusing the 800 nm pulses into a 2 mm-thick calcium fluoride plate. The WLC spanning 850–1200 nm was obtained by focusing the 800 nm pulses in a 3 mm-thick sapphire plate. The pump pulses were focused to a 425 μm diameter spot, while the probe pulses were focused to a 200 μm diameter spot. The excitation fluence was ≈700 μJ/cm$^2$. TA spectra were collected by using a fast optical multichannel analyzer working at the full 1 kHz laser repetition rate. The measured quantity is the differential transmission, $\Delta T/T$, from which the differential absorbance is calculated as: $\Delta A = -\Delta T/T$. Measurements were performed at room temperature in perfluorooctane solution in a 1 mm optical path quartz cuvette. For the global analysis (GA) of the obtained TA data, the open-source software Glotaran was used[55].

## Data availability

The data that support the findings of this study are available in the Supplementary Information file, and from the corresponding authors upon request. Crystallographic data can be obtained free of charge from the Cambridge Crystallographic Data Center via www.ccdc.cam.ac.uk/data_request/cif (CCDC no. 2045605).

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

## Acknowledgements

We acknowledge funding from the MIUR project NiFTy (PRIN2017, no. 2017MYBTXC). This work was supported by the Academy of Finland, Centers of Excellence Program (2014–2019; Grant No. 272361) and Academy Project (Grant No. 310799). We also acknowledge the provision of facilities and technical support by Aalto University at OtaNano - Nanomicroscopy Center (Aalto-NMC). Dr. Roberto Milani from VTT-Technical Research Center of Finland is acknowledged for his assistance in running HR-MALDI-TOF measurements.

## Author contributions

V.D. synthesized the thiol ligand. C.P., V.D., and I.D. performed nanoclusters synthesis and UV–Vis characterization. C.P. performed data analysis and wrote the manuscript. A.A. and F.Z. designed and performed computational studies. N.D. and G.T. performed X-ray studies. H.J. acquired STEM images. L.M., M.M., and G.C. designed and performed optical studies. C.P., V.D., R.H.A.R., F.B.B, P.M., and G.T. conceived the idea and designed the project. P.M., R.H.A.R., and F.B.B. supervised experimental activities. All the authors contributed to manuscript writing and revision.

## Competing interests

The authors declare no competing interests.
