## [Peer Review File · Nature Communications]

REVIEWER COMMENTS

Reviewer #1 (Remarks to the Author):

In this work, the authors report the synthesis and crystallization of an Au₂₅ nanocluster stabilized by fluoro-featuring thiols. They demonstrate that intracluster and intercluster F...F interaction are crucial to the crystallization. The design for the ligand is ingenious, and the demonstration for the proposal is clear. I would like to recommend publication of this work in Nature Communications after addressing the following issue:

1. The authors claimed that the F...F interaction on the cluster formed another protective layer for the cluster. I am thus curious about the stability of the cluster in comparison to Au₂₅ clusters stabilized by common thiols such as 2-phenylethanethiol (J. Am. Chem. Soc. 2008, 130, 5883).
2. The characterization of Au₂₅ is not complete. ¹H and ¹⁹F NMR are recommended for clarifying composition and purity of the cluster. The yield of the cluster should be provided too.
3. The ligand design is unique. Can more other clusters, especially those with different core structures be prepared?
4. Is this cluster luminescent? If so, it may find more potential applications.

Reviewer #2 (Remarks to the Author):

This is an interesting manuscript that describes a detailed series of studies on the new superfluorinated gold nanocluster [Au₂₅(SF₂₇)₁₈]. The cluster has been carefully characterised and its properties investigated. The manuscript is well written, with a few minor grammatical errors, and the authors have told a nice story.

The results are certainly noteworthy. While there are now a significant number of gold nanoclusters with the general formula [Au_(n)(L)_{(m)]_(q) in the literature and the majority of these have interesting physical and chemical properties, systems with highly fluorinated surfaces are less common. What makes this paper outstanding is the design feature starting with computational studies on the interactions in the F₂₇SH ligand that then led to the synthesis of the unique of the [Au₂₅(SF₂₇)₁₈] cluster, and the elegance with which the cluster has been isolated and its properties confirmed.}

What is less clear is how significant the study is and how it will influence the development of research in the area? The ability to self assemble nanoclusters is important and the use of F...F interactions has not been exploited significantly but is not unknown (as the authors point out), the solubility of the cluster in the fluoruous phase is significant. The work certainly complements the current literature and contributes to the area.

The results presented, and the logical design of the experiments, do provide a complete characterisation of the cluster. The authors have used a full range of techniques from mass spectrometry to microscopy, and rely on the single-crystal study for the complete structural characterisation. Crystallography of molecules of this size is challenging and the authors have done the best job possible with the quality of the data that could be obtained. The disorder within the ligand periphery and the effect of solvent in the crystal structure have been handled well. What is missing from the text is the inclusion of estimated standard deviations on the individual bond length reported in the text (where average values are quoted no esds are required).

To my understanding there are no significant flaws in the data analysis, interpretation or conclusions. The authors have not tried to over sell their results.

The methodology used in the synthesis and characterisation of the new cluster is sound as is the investigation of the properties of the cluster. Many aspects of the work are challenging but the combination of all the techniques used does allow firm conclusions to be drawn.

The level of detail in the methods section and in the supporting information is sufficient for the work to be reproduced.

Overall, the quality of the work presented is very high and my only question is how significant this study is in the general field of investigation? On balance, I feel that the work will be of interest to a wide range of chemists and material scientists, and I recommend publication subject to the minor corrections to the manuscript as mentioned above.

Reviewer #3 (Remarks to the Author):

This paper reports synthesis, structure determination and characterization of photophysical properties of highly-fluorinated Au nanoclusters [Au₂₅(SF₂₇)₁₈]₀. The main conclusion obtained by

experimental and computational studies is that both intracluster and intercluster F...F contacts play key roles in driving crystal packing and stabilization of $[\text{Au}_{25}(\text{SF}_{27})_{18}]_0$. Although these results are interesting, I think that the paper of the current form is more suitable to a more specialized journal. I believe that majority of the readers are more interested in unique effects of fluorinated ligands on the structures and properties of the Au nanoclusters as compared to those protected by the conventional thiolates. Actually, I can see many interesting differences between $\text{Au}_{25}(\text{SF}_{27})_{18}$ and $\text{Au}_{25}(\text{PET})_{18}$ (PET = phenylethanthiolate), such as color and fragmentation pattern (please see the detailed comments). I believe that the paper will become more attractive by highlighting the effects of fluorinated ligands on the electronic structures, atomic packing and thermal stability of the Au₁₃ core through the comparison between $\text{Au}_{25}(\text{SF}_{27})_{18}$ and $\text{Au}_{25}(\text{PET})_{18}$.

1. I do not agree with the statement that the optical spectrum in Figure 2c is assigned to neutral $[\text{Au}_{25}(\text{SR})_{18}]_0$. A hump at ~690 nm is characteristic to anionic $[\text{Au}_{25}(\text{SR})_{18}]^-$. Nevertheless, it is interesting that $\text{Au}_{25}(\text{SF}_{27})_{18}$ exhibits green color, indicating that the F₂₇S ligands affect the electronic structures of the Au₁₃ core.
2. The observation of the fragment $\text{Au}_{25}(\text{SF}_{27})_{12}$ is unique as compared to $\text{Au}_4(\text{PET})_4$ loss from $\text{Au}_{25}(\text{PET})_{18}$. How can the authors explain this ligand effect?
3. If the interligand F...F contacts is attractive, we can expect that $\text{Au}_{25}(\text{SF}_{27})_{18}$ exhibits higher thermal stability than $\text{Au}_{25}(\text{PET})_{18}$. It might also be interesting to see whether such interligand interaction affects the atomic structures by comparing the crystal structures of $\text{Au}_{25}(\text{SF}_{27})_{18}$ and $\text{Au}_{25}(\text{PET})_{18}$.
4. I do not understand why the authors described the target cluster as “20 kDa nanocluster” although the chemical formula was defined as $[\text{Au}_{25}(\text{SF}_{27})_{18}]_0$. Lowercase k should be used for molecular weight kDa.
5. The paper is difficult to read because the molecular formula and structure of F₂₇SH cannot be found in the main text but provided in SI. The authors should avoid using SF₂₇ in the abstract without the definition. I cannot find sulfur atoms in Figure 1a which shows single crystal of F₂₇SH.

POLITECNICO DI MILANO

DIPARTIMENTO DI CHIMICA, MATERIALI E INGEGNERIA CHIMICA "Giulio NATTA"

Prof. Pierangelo Metrangolo

Ph. +39-02-23993041, Fax: +39-02-23993180

E-mail: pierangelo.metrangolo@polimi.it

Milan, March 4, 2022

Point-by-point response to the reviewers' comments on manuscript:

“High-Resolution Crystal Structure of a 20 kDa Superfluorinated Gold Nanocluster”

by *Claudia Pigliacelli, Angela Acocella, Isabel Díez, Luca Moretti, Valentina Dichiarante, Nicola Demitri, Hua Jiang, Margherita Maiuri, Robin H.A. Ras, Francesca Baldelli Bombelli, Giulio Cerullo, Francesco Zerbetto, Pierangelo Metrangolo, and Giancarlo Terraneo.*

Reviewer 1

Comments:

1. The authors claimed that the F...F interaction on the cluster formed another protective layer for the cluster. I am thus curious about the stability of the cluster in comparison to Au₂₅ clusters stabilized by common thiols such as 2-phenylethanethiol (J. Am. Chem. Soc. 2008, 130, 5883).

We thank the reviewer for the valuable comment. The highly branched fluorinated structure of our cluster provides good chemical stability. Comparison to Au₂₅ clusters stabilized by more common thiols such as 2-phenylethanethiol is a good suggestion and will be performed in the continuation of this work.

2. The characterization of Au₂₅ is not complete. ¹H and ¹⁹F NMR are recommended for clarifying composition and purity of the cluster. The yield of the cluster should be provided too.

Following the reviewer's suggestion, we have provided yields, ¹H, and ¹⁹F NMR spectra of [Au₂₅SF₂₇]₁₈⁰ clusters, dissolved in perfluorooctane (see Supporting Information).

3. The ligand design is unique. Can more other clusters, especially those with different core structures be prepared?

We thank the reviewer for the valuable suggestion. We think that our ligand is highly versatile, and could be effectively used for stabilizing clusters with different core sizes. Further studies in this direction are currently undergoing.

4. Is this cluster luminescent? If so, it may find more potential applications.

We already reported in Chem. Comm. 2017, 621 that these clusters in bulk are luminescent. A detailed study on the luminescent properties of the isolated clusters in their reduced and oxidized states is still missing and will be the subject of further investigations. We thank the reviewer for the useful suggestion.

POLITECNICO DI MILANO

DIPARTIMENTO DI CHIMICA, MATERIALI E INGEGNERIA CHIMICA "Giulio NATTA"

Prof. Pierangelo Metrangolo

Ph. +39-02-23993041, Fax: +39-02-23993180

E-mail: pierangelo.metrangolo@polimi.it

Reviewer 2

Comments:

What is missing from the text is the inclusion of estimated standard deviations on the individual bond length reported in the text (where average values are quoted no esds are required).

Overall, the quality of the work presented is very high and my only question is how significant this study is in the general field of investigation?

We thank the reviewer for the positive comments and the suggestion about standard deviations of bond lengths. We have added their estimated values in the main text and in the Supporting Information. Concerning the impact of the work, as correctly highlighted by the reviewer, we think that our study provides significant insights on the role of F...F interactions in driving the self-assembly and crystallization of atomically precise clusters. Furthermore, the change in solubility resulting from oxidation of the gold core is unique and could be exploited in future phase switching catalytic applications.

Reviewer 3

Comments:

1. I do not agree with the statement that the optical spectrum in Figure 2c is assigned to neutral $[\text{Au}_{25}(\text{SF}_{27})_{18}]^0$. A hump at ~690 nm is characteristic to anionic $[\text{Au}_{25}(\text{SR})_{18}]^-$. Nevertheless, it is interesting that $\text{Au}_{25}(\text{SF}_{27})_{18}$ exhibits green color, indicating that the F_{27}S ligands affect the electronic structures of the Au_{13} core.

We thank the reviewer for the interesting observation. From previous literature reports (e.g., J. Phys. Chem. Lett. C 2008, 112, 14221 and Nanoscale 2015, 7, 1549), it is known that both neutral and anionic $[\text{Au}_{25}(\text{SR})_{18}]$ clusters show an absorption band between 650 and 700 nm, with a slight shift of its maximum according to the oxidation state of the core. This is actually visible also in the UV-vis spectra of our fluorinated species, before and after crystallization.

2. The observation of the fragment $\text{Au}_{25}(\text{SF}_{27})_{12}$ is unique as compared to $\text{Au}_4(\text{PET})_4$ loss from $\text{Au}_{25}(\text{PET})_{18}$. How can the authors explain this ligand effect?

The highly branched fluorinated structure of our cluster provides exceptional chemical stability to the Au_{25} core, which justifies that the fragment $\text{Au}_{25}(\text{SF}_{27})_{13}$ is formed leaving the Au_{25} core intact.

3. If the interligand F...F contacts is attractive, we can expect that $\text{Au}_{25}(\text{SF}_{27})_{18}$ exhibits higher thermal stability than $\text{Au}_{25}(\text{PET})_{18}$. It might also be interesting to see whether such interligand interaction affects the atomic structures by comparing the crystal structures of $\text{Au}_{25}(\text{SF}_{27})_{18}$ and $\text{Au}_{25}(\text{PET})_{18}$.

As commented above, the highly branched fluorinated structure of our cluster provides exceptional chemical stability to the Au_{25} core. However, its thermal stability compared to $\text{Au}_{25}(\text{PET})_{18}$ has not

POLITECNICO DI MILANO

DIPARTIMENTO DI CHIMICA, MATERIALI E INGEGNERIA CHIMICA "Giulio NATTA"

Prof. Pierangelo Metrangolo

Ph. +39-02-23993041, Fax: +39-02-23993180

E-mail: pierangelo.metrangolo@polimi.it

been studied, yet, and will be the subject of further investigations. We thank the reviewer for the useful suggestion.

4. I do not understand why the authors described the target cluster as “20 kDa nanocluster” although the chemical formula was defined as $[\text{Au}_{25}(\text{SF}_{27})_{18}]^0$. Lowercase k should be used for molecular weight kDa.

The exact molecular weight of our cluster is 20464.9 Da. However, since Nat. Comm. is wide interest journal, we preferred in the title to be less specific reporting only 20 kDa. We have corrected the mistake, replacing capital K with lowercase k.

5. The paper is difficult to read because the molecular formula and structure of F_{27}SH cannot be found in the main text but provided in SI. The authors should avoid using SF_{27} in the abstract without the definition. I cannot find sulfur atoms in Figure 1a which shows single crystal of F_{27}SH .

The molecular structure of thiol F_{27}SH is reported in Figure 1a. To better clarify the structure, highlighting sulfur atoms, we have added the color code of elements in the caption. We have also modified the abstract, specifying to what SF_{27} refers to.

Sincerely,
Pierangelo Metrangolo

REVIEWERS' COMMENTS

Reviewer #1 (Remarks to the Author):

I am satisfied with the authors' response and the revisions made in the revised manuscript. I can now recommend its publication.

Reviewer #2 (Remarks to the Author):

I have evaluated the modified manuscript and believe that the authors have satisfactorily addressed all the points raised by the reviewers. I have no further criticisms or comments on the manuscript. I think that the paper is of a very high quality and is suitable for publication in Nature Communications after technical editing.

Reviewer #3 (Remarks to the Author):

The paper has been revised properly according to the comments. I recommend publication of this paper in Nature Communication in the present form.